# Whole-Slide Images and Patches of Clear Cell Renal Cell Carcinoma Tissue Sections Counterstained with Hoechst 33342, CD3, and CD8 Using Multiple Immunofluorescence

Georg Wölflein [1,*,†] ![ID], In Hwa Um [2,†] ![ID], David J. Harrison [2,3] ![ID] and Ognjen Arandjelović [1] ![ID]

1   School of Computer Science, University of St Andrews, North Haugh, St Andrews KY16 9SX, Scotland, UK
2   School of Medicine, University of St Andrews, North Haugh, St Andrews KY16 9TF, Scotland, UK
3   Division of Laboratory Medicine, Lothian NHS University Hospitals, Edinburgh EH16 6SA, Scotland, UK
*   Correspondence: georg@woelflein.de
†   These authors contributed equally to this work.

**Abstract:** In recent years, there has been an increased effort to digitise whole-slide images of cancer tissue. This effort has opened up a range of new avenues for the application of deep learning in oncology. One such avenue is virtual staining, where a deep learning model is tasked with reproducing the appearance of stained tissue sections, conditioned on a different, often times less expensive, input stain. However, data to train such models in a supervised manner where the input and output stains are aligned on the same tissue sections are scarce. In this work, we introduce a dataset of ten whole-slide images of clear cell renal cell carcinoma tissue sections counterstained with Hoechst 33342, CD3, and CD8 using multiple immunofluorescence. We also provide a set of over 600,000 patches of size $256 \times 256$ pixels extracted from these images together with cell segmentation masks in a format amenable to training deep learning models. It is our hope that this dataset will be used to further the development of deep learning methods for digital pathology by serving as a dataset for comparing and benchmarking virtual staining models.

**Keywords:** cancer; digital pathology; machine learning; deep learning; computer vision; virtual staining; segmentation

## 1. Summary

With approximately 13,300 new cases every year, kidney cancer is the seventh-most-common type of cancer in the U.K. [1]. Clear cell renal cell carcinoma (ccRCC), a subtype of kidney cancer, whose name is derived from the appearance of its tumour cells under the microscope, is by far the most prevalent [2,3]. Studying its highly heterogeneous and vascularised tumour microenvironment (TME) is important for improving our understanding of the disease and its progression [4].

An important technique in clinical oncology and cancer research is the process of *immunostaining*, which facilitates the visualisation of various proteins in the cells of cancer tissue using artificial colouration [5] to distinguish between different cell types. Immunostaining assists pathologists in diagnosing cancer and deciding on treatment options [6–8]. Multiple immunofluorescence (mIF) allows different proteins to be visualised simultaneously by the enzymatic reaction between fluorescent-coated tyramide and horseradish peroxidase (HRP) [6,9,10]. In this work, we employed mIF with three different fluorophores to decorate ccRCC tissue sections for Hoechst 33342, cluster of differentiation 3 (CD3), and 29 cluster of differentiation 8 (CD8). The first is a widely used counterstaining fluorescent dye used to highlight cell nuclei [11], while the other two highlight specific cell subtypes: CD3 identifies T lymphocytes, and CD8 marks cytotoxic T lymphocytes.

Digitising whole slide images (WSIs) of tumour tissue as gigapixel images (typically around $100{,}000 \times 100{,}000$ pixels in size) has become an increasingly common practice in

the last decade, not only in research, but also clinical settings [12]. The contemporaneous advent of deep learning, which flourishes with the availability of large amounts of data, has sparked leaps in the computer vision community. These advancements, combined with the availability of digital pathology images, pave the way towards developing automated methods for WSI analysis. Potential applications vary from slide-level tasks such as patient risk stratification [13,14], to specific image tasks such as detecting cellular subtypes and their spatial distribution [15–17]. In this setting, deep learning not only has the potential to help reduce the workload of pathologists, but also to alleviate inter-observer bias, which is a common problem in pathology [18,19].

In an effort to facilitate deep learning research in digital pathology, we present a dataset of ten WSIs of ccRCC tissue, alongside the corresponding clinical data. The fact that our images contain three channels of information (Hoechst 33342, CD3, and CD8) makes our dataset particularly well-suited to the task of *virtual staining* [20], where a deep learning model is tasked with translating from one type of stain to another. In other words, given an image of stain *A*, the model should produce an image that appears as if the tissue section had instead been stained with another stain *B*. Our dataset, which is available in the BioImage Archive (http://www.ebi.ac.uk/bioimage-archive, accessed on 14 February 2023) under Accession Number S-BIAD605 [21], is presented in a manner that is suitable for training deep learning models by providing image patches and cell segmentation masks alongside the raw WSIs. Indeed, our dataset has already been used for training a modified generative adversarial network (GAN) [22,23] to convert Hoechst images to CD3 and CD8 [17]. Hoechst staining is significantly less expensive than CD3 and CD8 [17], so the ability to synthesise the former from the latter could also represent a significant costs saving.

## 2. Data Description

Our dataset consists of WSIs digitised from the tumour tissue of ten patients with ccRCC. The slides were sourced from the Pathology Archive in Lothian NHS (Ethics Reference 10/S1402/33). Using mIF, the slides were stained with Hoechst, CD3, and CD8 before being scanned at an objective of x40 on an Axioscan Zeiss scanner, resulting in a dataset of ten WSIs, each with three channels (Hoechst, CD3, and CD8).

We present the slides in two different formats: as raw WSIs and as preprocessed non-overlapping image patches of size 256 × 256 pixels covering the entire tissue region of the WSI. Furthermore, we provide the associated patients' clinical data in CSV format.

### 2.1. Raw Whole-Slide Images

We supply all ten WSIs in CZI format named according to the following convention: `ICAIRDXXX_MCM2FITC_CD3CY3_CD8CY5_MCK750.czi`, where `XXX` is the patient ID (referred to as the iCAIRD number in Section 2.3). As the naming convention suggests, the Hoechst intensities are captured in the FITC channel, CD3 in the CY3 channel, and CD8 in the CY5 channel. Figure 1 shows a low-resolution thumbnail of one of the WSIs.

### 2.2. Preprocessed Image Patches

The `patches.tar.gz` archive (70 G) contains the image patches. There are ten folders, one for each WSI, named according to the same convention as the raw WSIs in Section 2.1. For each patch, we supply a JSON file containing the metadata of the patch and the paths to the various image files associated with that patch. The JSON files are named `ICAIRDXXX_MCM2FITC_CD3CY3_CD8CY5_MCK750 [x=X, y=Y, w=256, h=256].json`, where `XXX` is the patient ID and `X`, `Y` give the location of the patch's top left edge in the WSI's pixel coordinates. Listing 1 explains the structure of the JSON file.

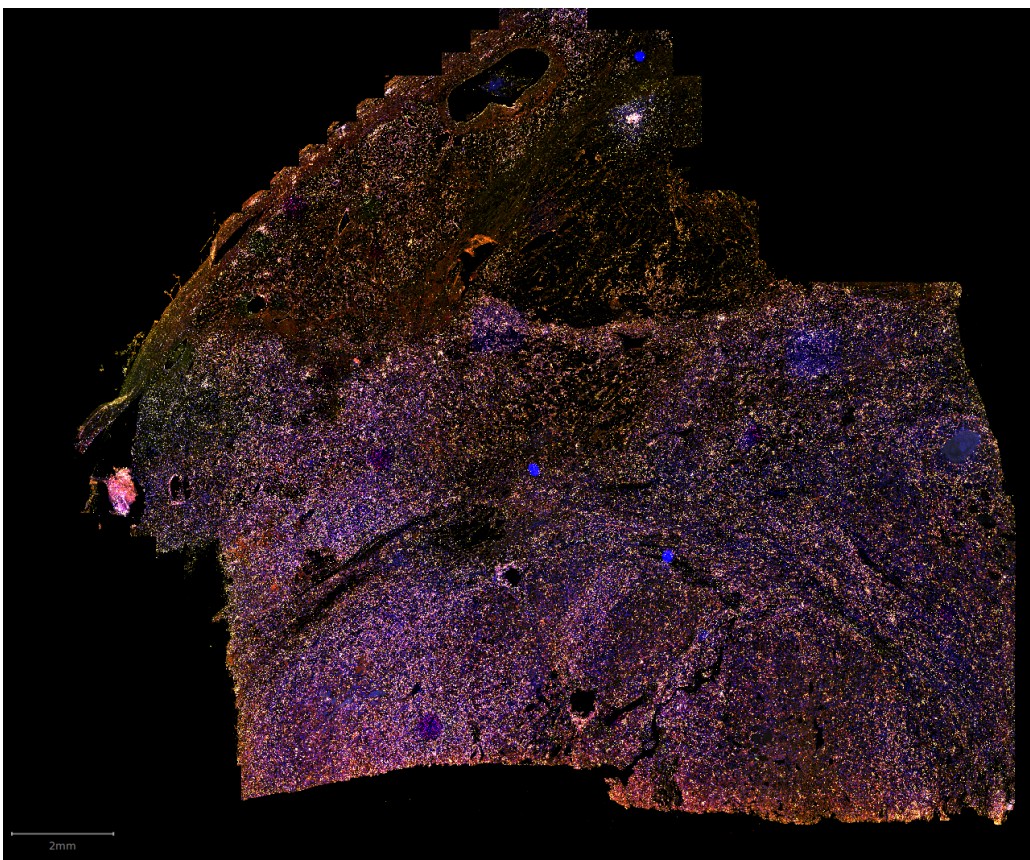

**Figure 1.** Thumbnail image of one of the WSIs in the dataset, displaying the Hoechst channel in blue, CD3 in yellow, and CD8 in red. Note that the individual cells are too small to be identified at the low resolution of this image.

**Listing 1.** Structure of the JSON file accompanying each patch.

```
{
"original_file": "ICAIRD1007_MCM2FITC_CD3CY3_CD8CY5_MCK750.czi",
"x": 51712,
"y": 51968,
"w": 256,
"h": 256,
"images": [
{
"file": "ICAIRD1007_MCM2FITC_CD3CY3_CD8CY5_MCK750 [...].png",
"mode": "mask",
"channel": "CD3"
},
// ...
]
}
```

In addition to the self-explanatory metadata fields referencing the original WSI file and patch coordinates, there is a field named `images`, which contains a list of image files associated with the patch. Each image file is described by a JSON object with the following fields: `file`, `mode`, and `channel`. The `file` field contains the name of the particular image file (located in the same folder as the JSON file itself). The `mode` field indicates the type of image file, which can be either `mask` (indicating a binary cell mask, i.e., a black and white image where white pixels represent the detected cells of a specific type) or `raw` (indicating a monochrome image with pixel intensities normalised according to Section 3.3.1). Table 1

lists the seven different image files associated with each patch alongside their respective `mode` and `channel` attributes. Each image is 256 × 256 pixels in size and supplied in PNG format.

**Table 1.** Types of image files associated with each patch, alongside their respective `mode` and `channel` attributes.

| Mode | Channel | Description |
|---|---|---|
| raw | H3342 | normalised Hoechst patch |
| raw | Cy3 | normalised CD3 patch |
| raw | Cy5 | normalised CD8 patch |
| mask | Hoechst | segmentation mask of all detected cells |
| mask | CD3 | segmentation mask of CD3+ cells (subset of Hoechst cells) |
| mask | CD8 | segmentation mask of CD8+ cells (subset of CD3+ cells) |
| mask | unclassified | segmentation mask of CD3- cells (subset of Hoechst cells) |

In total, the dataset consists of 627,519 non-overlapping patches. The 256 × 256 pixel patches under 20× magnification correspond to a physical size of about 58 × 58 μm. Statistics on the representation of each cell type in the dataset are provided in Table 2.

**Table 2.** Representation of cell subtypes across the dataset. Presence refers to the percentage of patches that contain at least one cell of the respective subtype. Area coverage means the percentage of pixels that are occupied by each cell subtype.

| | Hoechst | CD3 | CD8 |
|---|---|---|---|
| Total cells | 15,956,049 | 3,390,533 | 1,894,016 |
| Cells per patch | 25.42 | 5.40 | 3.02 |
| Presence | 99.95% | 93.08% | 71.61% |
| Area coverage | 26.48% | 05.01% | 03.02% |

*2.3. Clinical Data*

We provide a CSV file containing clinical data for the ten patients (`clinical_data.csv`, 571 B). The patients' iCAIRD numbers were used as the identifiers and match up with the names of the WSIs in Section 2.1 and the patches in Section 2.2. Data include the gender, age at surgery, five-year recurrence, and number of disease-free months after surgery. We also include morphological features assessed by a pathologist, including tumour size, lymph node involvement, and tumour grade, amongst others (Table 3 provides a full list of the columns).

**Table 3.** Columns in the clinical data table. Note that the "Disease-free months" column indicates a lower bound, as some patients may have experienced recurrence after the period of data collection.

| Column name | Format | Description |
|---|---|---|
| ICAIRD number | ICAIRD_XXX | patient ID |
| Gender | M or F | gender |
| Response | 0 or 1 | recurrence within 5 years after surgery |
| Age at surgery | whole number | age at surgery in years |
| Disease-free months | float | number of months with no recurrence |
| Fuhrman nuclear grade | 1 – 4 | Fuhrman grade [24] |
| ISUP nuclear grade | 1 – 4 | ISUP grade [3] |
| Tumour stage | 1a, 1b, 2a, 2b, 3a, 3b, 3c, or 4 | tumour size according to TNM system [25] |
| Tumour size | float | tumour size in cm |
| Node status | 0 or 1 | lymph node status according to TNM system [25] |
| Necrosis | 0 or 1 | whether necrosis is detected |
| Leibovich score (Fuhrman) | 0 – 11 | Leibovich score [26] using Fuhrman nuclear grade [24] |
| Leibovich score (ISUP) | 0 – 11 | Leibovich score [26] using ISUP nuclear grade [3] |

## 3. Methods

### 3.1. Multiplex Immunofluorescence Protocol

The method of staining the slides and obtaining the WSIs was described in the work of Wölflein et al. [17], but we include it here for completeness. The Leica BOND RX automated immunostainer (Leica Microsystems, Milton Keynes, U.K.) was utilised to perform mIF. The sections were dewaxed at 72 °C using BOND dewax solution (Leica, AR9222) and rehydrated in absolute alcohol and deionised water, respectively. The sections were treated with BOND epitope retrieval 1 (ER1) buffer (Leica, AR9961) for 20 min at 100 °C to unmask the epitopes. The endogenous peroxidase was blocked with peroxide block (Leica, DS9800), followed by serum-free protein block (Agilent, x090930-2). The sections were incubated with the first primary antibody (CD8, Agilent, M710301-2, 1:400 dilution) for 40 min at room temperature, followed by anti-mouse HRP conjugated secondary antibody (Agilent, K400111-2) for 40 min. Then, the CD8 antigen was visualised by Cy5-conjugated tyramide signal amplification (TSA) (Akoya Bioscience, NEL745001KT). Redundant antibodies, which were not covalently bound, were stripped off by ER1 buffer at 95 °C for 20 min. Then, the second primary antibody (CD3, Agilent, A045229-2, 1:400 dilution) was visualised by TSA Cy3, taking the same steps of the peroxide block to the ER1 buffer stripping of the first antibody visualisation. Cell nuclei were counterstained by Hoechst 33342 (Thermo Fisher, H3570, 1:100), and the sections were mounted with prolong gold antifade mountant (Thermo Fisher, P36930).

### 3.2. Whole-Slide Image Acquisition

The fluorescence images were captured using a Zeiss Axio Scan Z1 at an objective of x40 magnification. We used three different fluorescent channels (Hoechst 33342, Cy3, and Cy5) simultaneously to capture individual channel images under 20× object magnification with the respective exposure times of 10 ms, 20 ms, and 30 ms. Figure 2 shows the density curves of the three different channel intensities across the entire dataset.

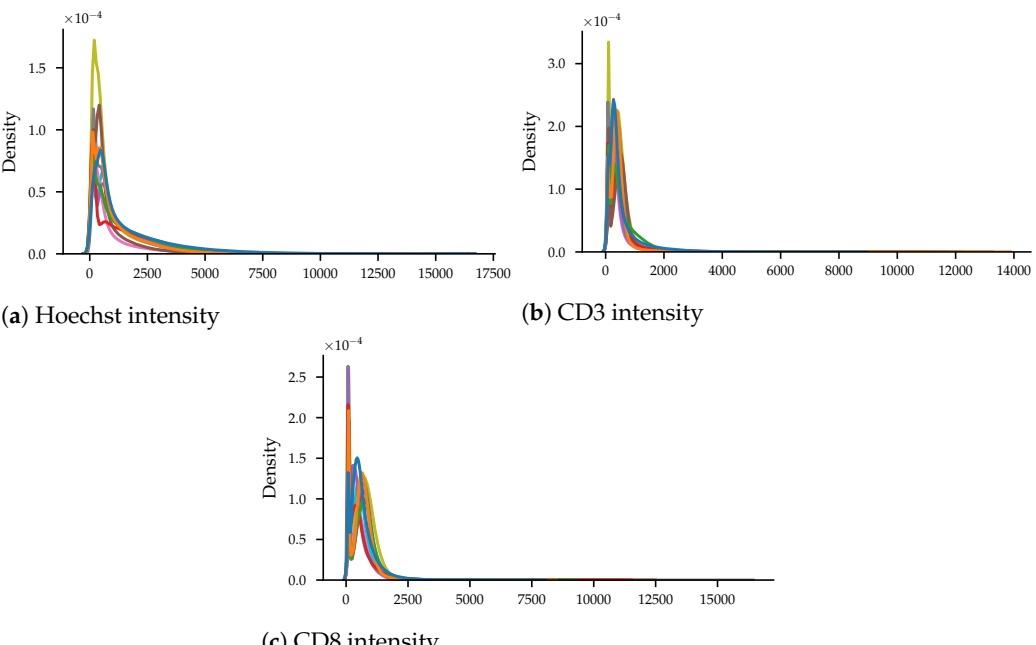

(**a**) Hoechst intensity     (**b**) CD3 intensity

(**c**) CD8 intensity

**Figure 2.** Intensity histograms of all 10 WSIs in the dataset (each WSI corresponds to a differently coloured line).

### 3.3. Patch Processing

#### 3.3.1. Intensity Normalisation

PNG files store pixels as 8-bit integers, which limits the dynamic range of the images. However, when examining the intensity histograms in Figure 3, we observed that most

pixel luminance was concentrated at the lower end of the range. A naïve quantisation of the image to the range $[0, 255]$ would lose most of the important information, specifically the variation at the lower end. To address this, we applied a form of thresholding.

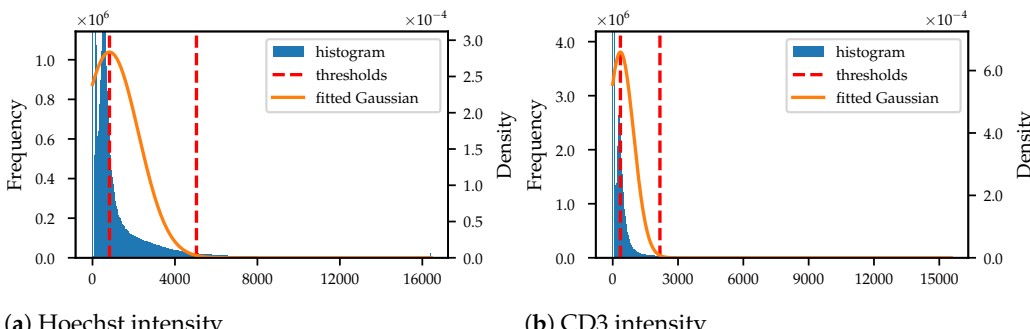

(**a**) Hoechst intensity    (**b**) CD3 intensity

**Figure 3.** Intensity histograms (left axes) and fit normal distributions (right axes) of a sample WSI's Hoechst and CD3 channels. The CD8 histograms behave similarly.

Each histogram in Figure 3 exhibits one main peak (disregarding the leftmost maximum at an intensity close to zero, corresponding to background pixels). Therefore, we found it sufficient to assume that the histogram follows a normal distribution $\mathcal{N}(\mu, \sigma^2)$, the parameters of which we obtained using maximum likelihood estimation. In practice, most of the important information is contained between the peak and three standard deviations to the right, i.e., in the range $[\mu, \mu + 3\sigma]$, indicated by the red lines in Figure 3. Eliminating intensities to the left of that peak ($x < \mu$) reduces the background noise. Moreover, pixels with high intensities ($x > \mu + 3\sigma$) are rare and can thus be discarded as well because they do not add much information. As a result, we transformed the intensities $x$ to the $[0, 1]$ range by the function:

$$f(x) = \min\left(1, \max\left(0, \frac{x - \mu}{3\sigma}\right)\right).$$

Note that we estimated the parameters $\mu$ and $\sigma$ derived from the histograms of the entire WSIs and not on a per-patch basis, due to the height variance between the patches. Furthermore, the described intensity normalisation procedure was applied to each stain separately, as illustrated by the sample patch in Figure 4.

### 3.3.2. Nucleus Segmentation

As indicated in Section 2.2, we supply the normalised image patches of each of the three channels (Hoechst, CD3, and CD8). However, we also include masks for each of the three channels (see Table 1), which are generated by a nucleus segmentation algorithm. These masks can be used to evaluate the quality of virtual staining algorithms [17,20] or even directly train segmentation models.

Our approach to nucleus segmentation uses the Hoechst channel as the starting point, instead of directly segmenting cells on the CD3/CD8 channels because those are less reliable. First, we segmented all nuclei in this channel using the StarDist algorithm [27], a popular deep-learning-based nucleus segmentation method. We employed StarDist because it is able to produce plausible non-overlapping masks even in crowded areas where instance segmentation models such as Mask-RCNN [28] tend to generate blobs of multiple cells [27]. This is because StarDist represents cells as star-convex polygons, whereas instance segmentation models simply operate on a pixel level. Figure 4g depicts the result of StarDist with a probability threshold of 0.6 and no cell expansion, as we employed it in our pipeline. Following Hoechst cell segmentation, we applied a threshold on the CD3 channel to identify which nuclei in the Hoechst mask were CD3$^+$ (Figure 4h). We repeated this process for the CD8 channel as well (Figure 4i). The entire nucleus segmentation pipeline (i.e., the aforementioned steps) was implemented as scripts using the QuPath software [29].

There were two factors that impacted the quality of the masks. First, Hoechst and CD3 stains may sometimes not align perfectly, which is evident in Figure 4e, where some of the high-intensity blobs do not match exactly with Figure 4d. This is because, while Hoechst stains the cell nuclei, CD3 is expressed only in a tiny part of a T cell's cytoplasm. Analogous reasoning applies to CD8. The second factor is the thickness of the slides (4 μm), which causes some cells to be out of focus, which becomes evident by the varying intensity levels in Figure 4a–c. As a result of both of these factors, there may be some cases where CD3[+] or CD8[+] cells may, by mistake, not be classified as such.

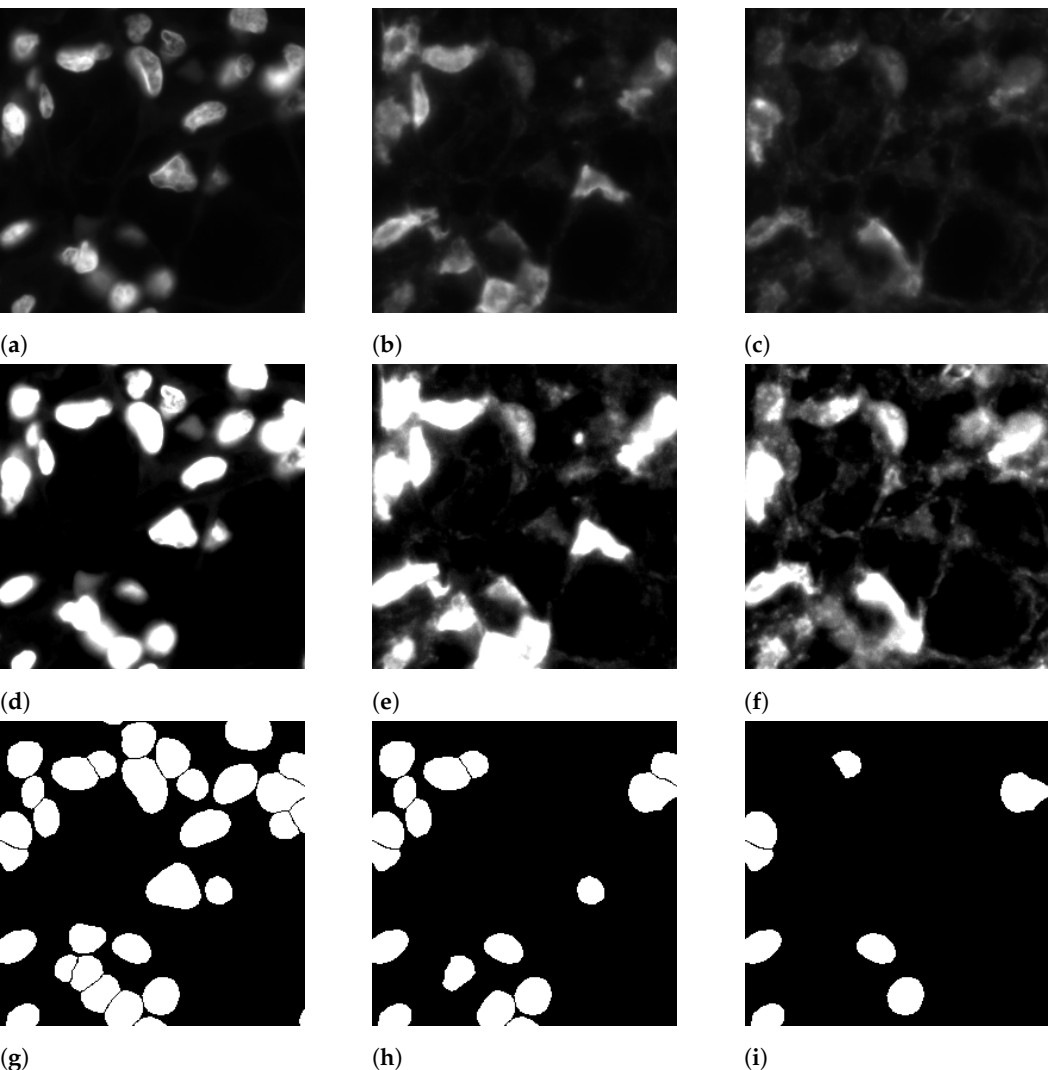

**Figure 4.** A 256 × 256 pixel patch extracted from the WSI in Figure 1, showing raw and normalised intensities for Hoechst, CD3, and CD8, as well as masks for different cell types. CD8[+] cells are a subset of CD3[+] cells because CD3 highlights all T cells, whereas CD8 binds only to cytotoxic T cells. (**a**) Hoechst. (**b**) CD3. (**c**) CD8. (**d**) normalised Hoechst. (**e**) normalised CD3. (**f**) normalised CD8. (**g**) StarDist [27] cell mask. (**h**) CD3[+] cells. (**i**) CD8[+] cells.

**Author Contributions:** Conceptualisation, G.W., I.H.U., D.J.H. and O.A.; methodology, G.W. and I.H.U.; software, G.W.; validation, G.W.; formal analysis, G.W.; investigation, I.H.U. and G.W.; resources, I.H.U. and D.J.H.; data curation, I.H.U. and G.W.; writing—original draft preparation, G.W.; writing—review and editing, I.H.U., D.J.H. and O.A.; visualisation, G.W.; supervision, O.A. and D.J.H.; project administration, O.A. and D.J.H.; funding acquisition, D.J.H. All authors have read and agreed to the published version of the manuscript.

**Funding:** G.W. is supported by Lothian NHS. This project received funding from the European Union's Horizon 2020 research and innovation programme under Grant Agreement No. 101017453 as part of the KATY project. This work was supported in part by the Industrial Centre for AI Research in Digital Diagnostics (iCAIRD), which is funded by Innovate UK on behalf of UK Research and Innovation (UKRI) (Project Number 104690).

**Institutional Review Board Statement:** The work was conducted in accordance with the Declaration of Helsinki and approved by the Ethics Committee of NHS Lothian NRS BioResource, REC-approved Research Tissue Bank (REC Approval Ref. 13/ES/0126, 3 February 2015).

**Informed Consent Statement:** Informed consent was obtained from all subjects involved in the study.

**Acknowledgments:** We would like to thank Craig Marshall, Lothian Biorepository, who granted access to the samples.

**Conflicts of Interest:** The funders had no role in the design of the study; in the collection, analyses, or interpretation of the data; in the writing of the manuscript; nor in the decision to publish the results.

## Abbreviations

The following abbreviations are used in this manuscript:

| | |
|---|---|
| ccRCC | clear cell renal cell carcinoma |
| TME | tumour microenvironment |
| mIF | multiplex immunofluorescence |
| IHC | immunohistochemistry |
| WSI | whole-slide image |
| GAN | generative adversarial network |
| CD3 | cluster of differentiation 3 |
| CD8 | cluster of differentiation 8 |
| TSA | tyramide signal amplification |
| HRP | horseradish peroxidase |
| JSON | JavaScript object notation |
| PNG | portable network graphics |
| CSV | comma-separated values |

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
