# Peer review of "Whole-Slide Images and Patches of Clear Cell Renal Cell Carcinoma Tissue Sections Counterstained with Hoechst 33342, CD3, and CD8 Using Multiple Immunofluorescence"

_data_

Round 1
Reviewer 1 Report
The authors described a dataset of ten whole slide images of clear cell renal cell carcinoma tissue sections counterstained with Hoechst 7 33342, CD3, and CD8 using multiple immunofluorescence. This dataset could be used for developing deep learning models for digital pathology by serving as a dataset for comparing and benchmarking virtual staining models. Datasets obtained through mIF are particularly useful for training virtual staining models, as it doesn’t require image co-registration. mIF allows different proteins to be visualized simultaneously from one single whole slide image at separate channels. Pixels that represent different proteins are already well aligned. The article was clearly and concisely written, and very easy to follow. There are only a few areas that need revision:
1. Page 7 Line 147, I would suggest adding more details about how StarDist was used to segment out cells on the Hoechst channel to increase the reproducibility of this study.
2. Page 7 Line 139, it might not be appropriate to say cell segmentation as StarDist segments nuclei given that Hoechst 33342 stain highlights cell nuclei.
3. Either design a metric to show the performance of StarDist nuclei segmentation on Hoechst 33342 images or providing literature evidence to support that StarDist would perform well.
4. It would be great if the authors could add some slide images from normal tissues. It is quite often that control tissues are needed to generalize models when training deep learning models.
5. I would suggest creating a chart to show density curves of Hoechst/CD3/CD8 intensity curves across all 10 slides. This would show us if there were any intensity heterozygosity across samples.
Reviewer 2 Report
It's very rare I have such an easy time reviewing a paper. Why? This paper is well written and clearly very useful to many researchers who would reuse this valuable public annotated data. I wish ALL public data sets had a nice writeup like this. Congrats.
Rather than describing the results of an experiment, the manuscript describes a large and highly annotated imaging data set being made public by the authors.
The manuscript describes data that includes WSIs from tumor samples of ten patients with ccRCC, stained with Hoechst, CD3, and CD8 and scanned at objective x40. The data is available as raw WSIs and as preprocessed non-overlapping image patches of size 256 × 256 pixels covering the entire tissue region, plus the patients’ clinical data in CSV format.
The availability of tiles and clinical data makes it immediately useful for people interested in machine learning and modeling.
The topic, a data set with annotations identifying T cells in histology images, is extremely important and somewhat rare. This kind of data is gold to people interested in using deep neural networks (AI) to build models that make automated cell type predictions.
Oftentimes, with public data, it's difficult to have a full grasp of where the data came from, how it was processed etc. But this manuscript makes that clear and much more feasible to reuse the data.
Again, the purpose of the manuscript is not describing an experiment, so there's no controls to speak of. They are describing data, for a journal called Data, and they do a great job of it.
The references are adequate, but the authors might consider adding https://www.frontiersin.org/articles/10.3389/fonc.2021.806603/full. But it's maybe not quite needed.
The figures and tables are fine... the authors could include a low resolution view of one of the WSIs, and maybe a tile (patch).
Author Response
We thank the reviewer for the assessment of our work. In line with the reviewer’s suggestions, we have added some more references in Section 1, including the article pointed out by the reviewer. Furthermore, we added a thumbnail image of one of the WSIs (Figure 1) which is where the sample patch in Figure 4 was extracted from.